# Ag-Contained Superabsorbent Curdlan–Chitosan Foams for Healing Wounds in a Type-2 Diabetic Mice Model

**DOI:** 10.3390/pharmaceutics14040724

**Published:** 2022-03-28

**Authors:** Elizaveta S. Permyakova, Anton S. Konopatsky, Konstantin I. Ershov, Ksenia I. Bakhareva, Natalya A. Sitnikova, Dmitry V. Shtansky, Anastasiya O. Solovieva, Anton M. Manakhov

**Affiliations:** 1Laboratory of Inorganic Nanomaterials, National University of Science and Technology “MISiS”, Leninsky Prospekt 4, 119049 Moscow, Russia; ankonopatsky@gmail.com (A.S.K.); shtansky@shs.misis.ru (D.V.S.); 2Laboratory of Pharmacological Active Compounds, Research Institute of Clinical and Experimental Lymphology—Branch of the ICG SB RAS, 2 Timakova Str., 630060 Novosibirsk, Russia; ershov_k@bk.ru (K.I.E.); xeniabahareva@yandex.ru (K.I.B.); sitnikovanat9@gmail.com (N.A.S.); solovey_ao@mail.ru (A.O.S.)

**Keywords:** superabsorbent dressings, diabetic wound regeneration, curdlan, chitosan, silver nanoparticles, XPS modelling

## Abstract

This study focused on the synthesis and characterization of pure curdlan–chitosan foams (CUR/CS), as well as foams containing Ag nanoparticles (CUR/CS/Ag), and their effect on the skin repair of diabetic mice (II type). The layer of antibacterial superabsorbent foam provides good oxygenation, prevents bacterial infection, and absorbs exudate, forming a soft gel (moist environment). These foams were prepared from a mixture of hydrolyzed curdlan and chitosan by lyophilization. To enhance the antibacterial properties, an AgNO_3_ solution was added to the curdlan/chitosan mixture during the polymerization and was then reduced by UV irradiation. The membranes were further investigated for their structure and composition using optical microscopy, scanning electron microscopy, energy-dispersive spectroscopy, FT-IR spectroscopy, and XPS analysis and modeling. In vivo tests demonstrated that CUR/CS/Ag significantly boosted the regeneration process compared with pure CUR/CS and the untreated control.

## 1. Introduction

The healing of burns and wounds is one of the most common health problems. Metabolic and physiological disorders (hypertension, malignancies, kidney disorders, diabetes, obesity, etc.) affect the normal process of skin reparation, resulting in ulcers, bedsores, and amputations, and causing death worldwide [1,2,3,4,5,6,7]. According to medical reports, 1–2% of the world’s population suffers from chronic wounds [8,9].

The exudate formed during wound healing is the body’s natural response to skin injury. To ensure successful healing, excessive exudate must be delayed, providing good oxygenation, a moist environment, and sterility [10].

Polysaccharides are widely used to prepare superabsorbent materials that can absorb and retain aqueous solutions hundreds of times their own dry weight [11,12]. Chitosan is a well-known carbohydrate polymer that has many potential clinical applications due to its antibacterial, anticoagulant, antitumor, and hemostatic properties [13]. β-glucans usually show a positive effect on the human immune system, providing antitumoral and antimicrobial effects [14]. Curdlan is a homopolysaccharide composed entirely of glucose monomers linked by β-1,3 glycosidic bonds. By varying the temperature, heating time, and curdan concentration, it is possible to obtain biomaterials of various strengths [15].

Polysaccharides can be formed in various morphological types of materials, such as film, hydrocolloid, hydrogel, fibers or foam, which significantly affects their properties [16]. Porosity has been reported to increase with increasing the absorption capacity and absorption rate [17].

For example, the curdlan/chitosan biomaterials were obtained by Wang et al. [18] and Przekora et al. [19] by mixing polysaccharides in a ratio of 2:1. However, the preparation methods and the structures of dressings were different: membrane (evaporation at 60 °C [18]) and highly porous (66–77%) foam (lyophilization process [19]). A comparison of the absorbance capacities demonstrated that foam can absorb liquids >40 times more efficiently than membranes. The swelling ability of curdlan/chitosan (1:2) electrospun nanofibers (d = 216 ± 60 nm) was 350% [20].

Although chitosan and curdlan derivatives have antibacterial properties, this is not enough to provide 100% antimicrobial protection. A promising and widely used approach is the addition of Ag nanoparticles to enhance the antibacterial effect [21].

Yu et al. [22] produced a gelatin/chitosan composite containing Ag NPs (0, 1, 3, and 5%) by lyophilization. They demonstrated the enhanced antibacterial activity of the obtained Ag-loaded materials against *E. coli* and *S. areus* cells, and found a directly proportional increase in the porosity and water absorbance capacity depending on the concentration Ag NPs. This was due to an increase in the viscosity of the solution and an increase in the formation of bubbles during stirring. Thus, more pores are formed during lyophilized.

In this study, novel CUR/CS/Ag NPs foams were developed by polymerization at 90 °C with dropwise addition of an AgNO_3_ solution followed by UV irradiation. It was demonstrated that the presence of Ag significantly affects the swelling rate. In vivo tests in mice with genetically determined type 2 diabetes mellitus revealed an enhanced effect of CUR/CS/Ag foams on skin repair compared to pure CUR/CS and the untreated control. Thus, the proposed facile strategy for the production of biocompatible superabsorbent foams opens up promising prospects for the creation of new functional platforms for temporary skin substitutes for the healing and regeneration of chronic wounds.

## 2. Materials and Methods

### 2.1. Preparation of Foam-like Curdlan–Chitosan (CUR/CS) and Curdlan–Chitosan-Ag NPs (CUR/CS/Ag) Biomaterials

Curdlan/chitosan foams were prepared as described elsewhere [19]. Briefly, aqueous solutions containing 2 wt.% of curdlan (99%, Qingdao SIgma Chemical, Qingdao, China) and 1 wt.% of chitosan (99%, Mw 100 kDa, Qingdao SIgma Chemical, Qingdao, China) in 1% (*v*/*v*) acetic acid solution were mixed (1:1) and preheated to 55 °C with continuous stirring with a magnetic stirrer. Then, the obtained mixture was transferred into a round-bottom flask, which was placed in a glycerin bath at 90 °C for 20 min.

Curdlan/chitosan/Ag membranes were prepared by the reduction of AgNO_3_ (Alfa Aesar, A Johnson Matthey Company, Tewksbury, MA, USA) under ultraviolet (UV) irradiation. Then, 0.01 N AgNO_3_ (≈0.5 wt.% Ag in terms of the dry residue of the mixture) was added dropwise to the curdlan/chitosan solution during the polymerization reaction (20 min; 90 °C; V = 60 mL/h), after which the solution was irradiated with a UV lamp (wavelength λ = 185 nm). The same H_2_O volume was added to the pure curdlan/chitosan solution to compensate for the volume difference. Finally, the resultant samples were cooled, frozen at −196 °C, and subjected to a lyophilization process (Martin Christ Alpha 1-2 L.D. plus, Osterode am Harz, Germany) for 24 h to obtain a foam-like structure. A schematic of the curdlan/chitosan and curdlan/chitosan/Ag foam fabrications is shown in Figure 1.

### 2.2. Characterization

The sample morphology was examined by optical and scanning electron microscopy. Optical analysis was performed using a BX51 optical microscope (Olympus, Tokyo, Japan). SEM analysis was carried out on a JSMF 7600 microscope (JEOL Ltd., Tokyo, Japan) equipped with an energy-dispersive X-ray spectrometer. The samples were coated with a ~5 nm thick Pt layer to compensate for the surface charge and to prevent sample damaging. 

The sample chemical characterization was performed by XPS, energy-dispersive X-ray spectroscopy (EDXS), and FTIR spectroscopy. FTIR spectra (100 scans) were recorded with a step of 4 cm^−1^ on a Vertex 80v FTIR spectrophotometer (Bruker, Billerica, MA, USA) with a parallel beam transmittance accessory. The spectra were collected at room temperature. The XPS method was used to determine the surface chemical composition using an Axis Supra spectrometer (Kratos Analytical, Manchester, UK). The maximum lateral dimension of the analyzed area was 0.7 mm. The spectra were fitted using the CasaXPS software after Shirley-type background subtraction. The binding energies (BE) for all carbon and oxygen environments were taken from the literature [23,24,25]. The BE scale was calibrated by setting the CH_x_ component at 285 eV.

### 2.3. Water Absorbance

Dry biomaterials weighing 10 mg ± 1 mg (8 mm × 8 mm, 2 mm in height) were immersed in PBS at 37 °C. After sufficient swelling, the biomaterials were removed from PBS, and excess water on their surfaces was removed with filter paper until a constant sample weight was fixed. The water absorption (hereafter denoted as WA%) of curdlan/chitosan foams was calculated as W_A_ = (W_w_ − W_d_)/W_d_ × 100%, where W_w_ and W_d_ are the weights of the wet and dry foams, respectively.

### 2.4. In Vivo Assay

The Ethics Committee approved the procedures of the RICEL-branch of ICG SB RAS (№ 170 dated 22 January 2022). All animal procedures were carried out in accordance with the protocols approved by the Bio-ethics committee of the Siberian Branch of the Russian Academy of Sciences, recommendations for the proper use and care of laboratory animals (European Communities Council Directive 86/609/C.E.E.), and the principles of the Declaration of Helsinki. 

B.K.S.Cg Dock7 <m>+/+Lepr <db>/J mice (were obtained from SPF vivarium of the Institute of Cytology and Genetics SB RAS, Novosibirsk, Russia, further denoted as db/db) (female, bodyweight: ≈30–40 g, 5 month old) were randomly divided into two groups of curdlan/chitosan and curdlan/chitosan/Ag (n = 3). Mice were anesthetized with 35 mg/kg zoletil (Valdepharm, France) and 7 mg/kg ksilazin (Interchemie werken «De Adelaar» BV, Holland). Then, the backs were shaved and two 1 cm × 1 cm full-thickness cutaneous defects were made on each side of the spine (one is the control (untreated) and the second is the treated material). The defects were covered with either curdlan/chitosan or curdlan/chitosan/Ag for 10 days. After that, the wounds were photographed until one of the wounds healed (24 days). The mice were then sacrificed and the wound tissues were collected and immersed in 4% formaldehyde for hematoxylin and eosin (H&E) staining and were photographed with a microscope (Zeiss Axio observer Z1, Oberkochen, Germany). Defect areas were photographed with a Canon camera and measured with the Image J program. To calibrate the magnification of photographs, a reference square of 1 cm × 1 cm in size was placed in the wound area. Wound areas were determined by counting the surface area. 

## 3. Results

### 3.1. Fabrication of CUR/CS and CUR/CS/Ag Foams and Their Structural Analysis

It is known that a slightly acidic environment, which can be provided with appropriate dressings, promotes the regeneration of chronic wounds by stimulating fibroblast proliferation, preventing bacterial contamination, and decreasing protease activity [26]. The use of acetic acid ensures the solubility of chitosan derivatives and makes it possible to achieve a suitable pH (pH of the polymerizable mixture of 5.9) to stimulate the repair process. Thus, using acetic acid provided the solubility of chitosan-derivatives and allowed for achieving a suitable pH (the pH of polymerized mixture 5.9) to stimulate the reparation process.

The structure of curdlan and chitosan and the possible chemical interactions between their functional groups (hydroxyls, hydrogen, and amino-groups are involved in the formation of hydrogen bonds) are presented in Figure 2. 

The addition of AgNO_3_ to a polymer solution is a widely used method for preparing composite materials containing Ag NPs. This approach provides a uniform distribution of Ag NPs in the resulting material, and allows one to control the concentration of Ag NPs.

To obtain a foamy microstructure of both biomaterials (CUR/CS and CUR/CS/Ag) with a superabsorbent capacity, a lyophilization method was used. Figure 3 shows images of the fabricated biomaterials obtained using optical and scanning electron microscopy, which clearly show the porous structure of both samples. Interestingly, CUR/CS/Ag has a foamy structure with a higher microporositiy than CUR/CS, which can be explained by the increased viscosity of the initial Ag-contained solution. The foamy structure should provide a good absorption capacity and appropriate oxygenation for wound regeneration. The chemical composition of the obtained materials, determined by the EDXS method, is given in Table 1.

### 3.2. XPS Analysis and Modelling of CUR/CS and CUR/CS/Ag Biomaterials

The sample surfaces were analyzed by XPS and the atomic compositions are reported in Table 2. It can be seen that the surface compositions have some differences from the EDXS results (Table 1), which is most likely due to differences in the depth of analysis: ~10 nm (XPS) and ~1000 nm (EDXS). A significant difference in atomic compositions can also be seen when comparing CS, CUR, and CUR/CS/Ag samples. CS shows 8.8 at.% nitrogen, while CUR reveals no nitrogen and a higher oxygen content. CUR/CS/Ag exhibits a lower nitrogen content than pure CS (due to mixing with CUR) and 0.4 at.% Ag.

In order to analyze the material structure in more detail, the high resolution XPS spectra of C1s, N1s, and Ag3d were analyzed.

The XPS C1s signal of the pristine chitosan powder (CS) was fitted with a sum of four components: CH_x_ (BE = 285.0 eV), C–N (285.8 eV), C–O (286.9 eV), and N–C=O (288.4 eV) (Figure 4). The full width at half-maximum was set to 1.2 eV for all components, and the line shape was a mixture of 30% of Lorentzian and 70% Gaussian. This line shape is the same for all components, except the signal fitting of the CS/CUR/Ag C1s signal. The functional composition of the CS surface is in good agreement with the structural scheme of chitosan, which consists of N-acetyl glucosamine and glucosamine. The N–C=O and CH_x_ components are attributed solely to N-acetyl glucosamine units, while the C–N component (amine groups) are ascribed to glucosamine. The O1s signal fitting of CS was performed with a single C–O component centered at 533.3 eV with a FWHM of 1.5 eV (Figure 5a). The N1s signal of CS was very informative, as two distinct nitrogen peaks were observed: protonated amines NH_3_^+^ (BE = 403.2, FWHM = 1.2 eV) and amides N–C=O (BE = 399.9 eV, FWHM = 1.2 eV). Using the NH_3_^+^/N–C=O concentration ratio (Figure 6), the ratio of N-acetyl glucosamine to glucosamine units was determined to be 1:1.2, i.e., glucosamine units were dominating.

The pristine curdlan (CUR) sample only had oxygen and carbon on its surface. The fitting of the C1s signal (Figure 4b) was performed with the sum of three components: CHx (BE = 285 eV, FWHM = 1.3 eV), C–O (BE = 286.5 eV, FWHM = 1.1 eV), and C=O (BE = 287.9 eV, FWHM = 1.8 eV). The CH_x_ and C=O contributions were not expected to be found on the curdlan surface, as they were absent in the curdlan structure. The presence of these components may be associated with surface contaminations. The O1s signal of CUR also revealed the C–O (BE = 533.0 eV, FWHM = 1.5 eV) and C=O (BE = 531.1 eV, FWHM = 1.5 eV) components (Figure 5b). However, the concentration of C=O was very low.

The carbon, oxygen, and nitrogen environments were also highly interesting in the detailed analysis. The CUR/CS/Ag samples impregnated with Ag nanoparticles showed an Ag signal in both the EDXS and XPS spectra. The Ag content was estimated to be ~0.4 at %. The presence of silver in the CUR/CS/Ag sample was evidenced by the presence of the Ag3d XPS peak (Figure 6c) at position 367.8 eV, corresponding to the Ag_2_O phase. The XPS C1s signal of CUR/CS/Ag was approximated using the CASA XPS software by introducing a new line that replicated the form of the signal from the CS and CUR samples. Finally, the C1s signal was fitted with a sum of three components: CS, CUR, and CHx (BE = 284.9 eV, FWHM = 0.8 eV). The intensity of the CHx component was only 6% and was probably associated with the surface contamination. The CS/CUR ratio, estimated by the curve fitting, was 1:1.14, with a slight predominance of curdlan. A similar ratio was estimated from the elemental composition of the CS/CUR-Ag sample. Indeed, as nitrogen is present only in chitosan, and we know the atomic composition of CS, the CS/CUR ratio can be estimated using Equation (1), where [C]_CS_, [N]_CS_, [C]_CUR/CS/Ag_, and [N]_CUR/CS/Ag_ are carbon and nitrogen concentrations in the CS and CUR/CS/Ag samples, respectively.
(1)$$\mathrm{CS}/\mathrm{CUR}=\left[1-\frac{{\left[C\right]}_{\mathrm{CS}}\times {\left[N\right]}_{\mathrm{CUR}/\mathrm{CS}/\mathrm{Ag}}}{{\left[C\right]}_{\mathrm{CUR}/\mathrm{CS}/\mathrm{Ag}}\times {\left[N\right]}_{\mathrm{CS}}}\right]:\left[\frac{{\left[C\right]}_{\mathrm{CS}}\times {\left[N\right]}_{\mathrm{CUR}/\mathrm{CS}/\mathrm{Ag}}}{{\left[C\right]}_{\mathrm{CUR}/\mathrm{CS}/\mathrm{Ag}}\times {\left[N\right]}_{\mathrm{CS}}}\right]$$

According to this equation, the CS/CUR ratio is 1:1.15. The XPS N1s spectrum of the CUR/CS/Ag sample shows that the percentage of glucosamine and N-acetyl glucosamine is the same. 

### 3.3. FT-IR Analysis of CUR/CS and CUR/CS/Ag Biomaterials

The FT-IR spectra of the pure components (chitosan and curdlan) and the spectra of hybrid chitosan/curdlan foams and Ag-loaded foams (cross-linked at 90 °C) are presented in Figure 7. To interpret the obtained peaks, it is necessary to analyze the structure of the polysaccharides and the possible interactions between them (hydroxyls, hydrogen, and amino-groups are involved in the formation of hydrogen bonds (Figure 2).

The as-prepared biomaterials contain two types of polysaccharides: chitosan (CS) with N-acetyl glucosamine and glucosamine units and curdlan (CUR) built from glucose units using β-(1,3)-glucan linkages. All samples contain glycosidic C–O bonds (1027 cm^−1^), C–O–C bonds in the ring (1065 cm^−1^), C–C and C–O bonds (992 cm^−1^), and aliphatic groups (region 3000–2800 cm^−1^). The broad complex band at 3700–3000 cm^−1^ can be associated with the OH– (≈3300) and CONH–groups (maxima at 3364 and 3290 cm^−1^) [26]. Indeed, the intensity of the spectrum of the CS and CUR/CS samples containing N-acetyl-glucosamine units in this region is higher than that of the CUR counterpart. The specific absorption bands at 890, 1080, and 1160 cm^−1^ indicate the presence of β-(1,3)-glucan linkages in curdlan. 

The CS spectrum has specific nitrogen-associated bands at 1214 cm^−1^ (C-N stretch) and 1633 cm^−1^ (amide I of β-pleated sheet structures) [27]. Peaks in the region of 1480–1444 cm^−1^ can be associated with CH and NH of amide II and aliphatic CH deformation [28]. A maximum at 915 cm^−1^ was assigned to the C–C, O–C, C–O, and C–CH_3_ deformations [29].

In the CUR/CS sample, the band at 3085 cm^−1^, associated with -NH groups in chitosan, is not observed. This could be caused by hydrogen bonds between C=O⋯HN species in the N-acetyl-glucosamine units of chitosan and 1,3-β-D-glucan units [30].

### 3.4. PBS Absorbance Ability of CUR/CS and CUR/CS/Ag Biomaterials

The wetting time of the superabsorbent foam dressings was measured, and the obtained results are shown in Figure 8. Shorter wetting times indicate better wetting. As can be seen, the addition of Ag NPs significantly affects the absorption rate. CUR/CS/Ag foams had a much faster wetting rate than the foam dressings without Ag NPs (6 s for CUR/CS/Ag versus 60 s for CUR/CS). However, the amount of absorbed liquid was almost the same.

### 3.5. In Vivo Assay

The regenerative potential of the developed materials was evaluated using the model of a full-thickness skin wound. Mice with genetically determined type 2 diabetes mellitus were used. This disease is characterized by a frequent complication of chronic long-term non-healing wounds, the therapy for which is difficult due to impaired perfusion and innervation in such patients. Therefore, the search and development of new effective materials is an urgent task. As shown in Figure 9 and Figure 10 untreated wounds in db/db mice heal very slowly, so no healing was achieved during the 30-day observation. The use of CUR/CS led to a decrease in the size of wounds; however, complete healing was not observed. At the same time, it was found that the use of CUR/CS/Ag foam significantly accelerated healing already in the first 10 days after wounding (Figure 9). 

The histology analysis (Figure 11) of the wound after 24 days revealed an accumulation of inflammatory cells (polymorphonuclear and mononuclear) in the untreated controls (Figure 11). The wounds are covered with scabs, filled with a mixture of non-cellular debris and dead leukocytes, and re-epithelialization is low. Along the edges of the wound, neuroepithelial growths are clearly visible, the border between the wound and areas of healthy skin are noticeable. Moderate re-epithelialization is observed in the wound treated with CUR/CS, covering a partially organized layer of medium-thick granulation tissue with signs of pericellular edema and, accordingly, sparse cell density. Capillary neovascularization is noted. There are remnants of a scab of non-cellular remnants of red blood cells and exudate. In the CUR/CS/Ag treated specimen, the wound defect is no longer present. There is a layer of granulation tissue formed during the remodeling process. Vessels that are newly formed capillaries, as well as vessels of larger diameter, are also observed. There are no signs of edema and active inflammatory reaction. Re-epithelialization is almost complete, but in places, detachment of the epithelium is noted. The remains of a scab and crusts were not found. Thus, the observed histological picture confirms the almost complete healing of the wound during CUR/CS/Ag therapy.

## 4. Discussion and Final Remarks

Although silver-containing wound products like 1% cream silver sulfadiazine or Ag-contained dressings (AQUACEL^®^ Ag Extra™ dressings, ConvaTec; Silver Alginate, Areza Medical;and ATRAUMAN AG, Hartmann, etc.) are widely used in diabetic wound healing applications and demonstrate good results, the toxicity of Ag ions and Ag NPs is still a hot topic in the scientific field. Some articles [31,32,33] have reported that the mechanisms of toxicity with Ag NPs included involve oxidative stress, genotoxicity, activation of lysosomal activity, disruption of the actin cytoskeleton and stimulation of phagocytosis, an increase of MXR transport activity, and an inhibition of Na-K-ATPase. However, the main contribution in the toxicity of Ag NPs was determined as takeover Ag NPs by endothelial cells, and induced concentration-dependent intracellular ROS elevation [34]. It was shown that covering Ag NPs with polymer or biomolecules (polyvinylpyrrolidone [35,36], citrate [35,37], tyrosine [33], PEG [36], etc.) decreased the cytotoxicity. In this research, Ag NPs incorporated into the structure of the CUR/CS/Ag material, where antibacterial polymers (curdlan and chitosan) decreased the required concentration of Ag NPs and most probably decreased the cytotoxicity of Ag NPs due to binding the organic molecules to the metal surface through nucleophilic functional groups. On the other hand, the rapid release of Ag ions can promote the activation of specific immunocytes. In particular, several studies have demonstrated that AgNPs can directly activate the innate immune response, especially macrophages [38,39].

Table 3 summarizes some recent results aimed at the development of new dressings for wound healing. It should be noted that the regeneration rate depends on the type (burn or wound) and size of the skin disruption, and also on the inflammation processes or diabetic type of wound. In addition to silver nanoparticles, dressings can also include nanoparticles of ZnO [40,41], Cu [12,20], Fe_3_O_4_ [42,43], and Ca [44].

Clinic research [45] has reported about the influence of copper oxide microparticles (CuO MPs) on the healing of diabetic foot ulcers (13 patients). The researchers soaked standard of care dressings in CuO MPs solutions and then evaluated the differences in the closure rates. It was shown that CuO MPs stimulate the healing of non-infected stagnated wounds in diabetic patients. 

In another clinical article, dressings based on chitosan hydrogel loaded with Ag NPs and calendula extract were tested to heal chronic wounds (two patients) [46]. Using these dressings (changed every 7 days, over 2 weeks) considerably decreased pain and inflammation, until the symptoms were eliminated. At the end of the 15 days of treatment, it was observed that they had adapted to the size of the wound and remained completely adhered, so it was decided to leave them for another period of 15 days. After this period, it was observed that the wounds had healed. 

More recent research has also demonstrated the efficiency of using bioactive molecules such as growth factors [47,48], enzyme [49], or receptors [50] in the composition of dressings to enhance the therapy of chronic wounds.

**Table 3 pharmaceutics-14-00724-t003:** Recent studies of the developed dressings for wound healing.

№	Dressings	Size of Wound	Observations In Vivo	Animal	Reference
1	Structure: nanofibers had a diameter between 200 to 300 nm, size of NPs 50–100 nm Composition: chitosan/polyvinyl alcohol/copper NPs	Wound: 1.5 cm × 1.5 cm Area: 225 mm^2^	The wound closure rate of the negative control group was 18.46%, 59.89%, 62.42%, and 88.07%, and the wound closure rate of the positive control group was 25.33%, 72.85%, 95.32%, and 97.90% for 3, 7, 11, and 15 days, respectively	Rat	[42]
2	Structure: film and gel functionalized by NPs (11.5–18.71 nm) Composition: bacterial cellulose/ betulin diphosphate/ ZnO NPs	Burn rea: 1400 ± 50 mm^2^ Depth: 3–5 mm	On day 21, the wound area treated with BC-ZnO NPs-BDP films was reduced by 34.3%, while when treated with ZnO NPs-BDP oleogel, a large decrease of up to 40.6% was observed. In the untreated control, the closure rate was just 19.2%	Rat	[40]
3	Structure: electrospun fibers (648.1 ± 72.2 nm) with NPs Composition: PLA + Ca NPs	12 mm square skin wounds Area: 452 mm^2^	80% contraction in wound area vs. 62% in the untreated control on 8 days	Diabetic mice	[44]
4	Structure: hydrogel with NPs (size 99.1 ± 2.3 nm) Composition: chitosan/PEG/Ag NPs	20 mm square skin wounds Area: 1256 mm^2^	A 47.7 ± 1.8% contraction in the wound area was recorded with the AgNPs impregnated chitosan-PEG hydrogel group, compared to 12.6 ± 1.3% in the negative control	Diabetic rabbit	[51]
5	Standard of care dressings impregnated with copper oxide microparticles (COD)	9.26 ± 6.9 cm^2^ (range of 1.35–23.6 cm^2^)	Following 1 month of copper improved treatment, there was a clear reduction in the mean wound area (53.2%; *p* = 0.003), an increase in granulation tissue (43.37; *p* < 0.001), and a reduction in fibrins (47.8%; *p* = 0.002). In the control group, wound closure was less than 20%	Clinic diabetic foot ucler	[45]
6	Structure: hydrogel Composition: gelatin/hyaluronic acid/thrombomodulin	8-mm diameter round-shaped wound Area: 201 mm^2^	On day 10, wound closure was 80% for hydrogel with thrombomodulin vs. the 40% untreated control	Mice	[50]
7	Structure: electrospun fibers (90–120 nm) Composition:Enteromorpha polysaccharide and polyvinyl alcohol (PVA)	10-mm diameter round-shaped wound Area: 314 mm^2^	On day 9, the wound contraction rate for the PVA/EPP1 group reached nearly 72% vs. 54% for the control group	Diabetic mice	[52]
8	Structure: electrospun fibers (110 ± 74 nm) Composition: hydroxypropyl methylcellulose (HPMC)/polyethylene oxide (PEO)/ Beta-glucan	1 cm × 1 cm Area: 100 mm^2^	βG-nanofibers 95% healing vs. 40% healing of control in 24 days	Diabetic mice	[53]
9	Structure: hydrogel Composition: chitosan, heparin and poly (γ-glutamic acid) and loaded with superoxide dismutase	10-mm diameter round-shaped wound Area: 314 mm^2^	After 21 days, closure rate is 92.0% ± 3.7% compared with the control group (85.4% ± 2.4%)	Diabetic mice	[49]
10	Structure: hydrogel Composition: glycol chitosan, loaded by growth factors (VEGF and PDGF-BB)	5-mm diameter round-shaped wound Area: 157 mm^2^	On day 3, hydrogel dressing demonstrated 60% closure rate vs. less than 5% for the Duoderm dressing	Diabetic mice	[48]

The present study examined the ability of Ag-contained CUR/CS foams to accelerate the wound healing in diabetic mice. It has been shown that Ag NPs in CUR/CS foams significantly affect the PBS absorbance capacity (the swelling rate increases by ~ten times) and accelerate the healing process of the diabetic wound.

The wounds treated with CUR/CS/Ag showed complete healing after 24 days, which was accompanied by the formation of new vessels of different diameters and significant re-epithelialization. At the same time, CUR/CS-induced healing was only 20%, and the control showed an increase in the area of the wound by 20+%, which is associated with inflammatory processes. It is possible that such a difference in effect is associated with different material dissolution rates. The developed CUR/CS/Ag foams could meet the needs for clinical practice, and may have future medical applications for wound care especially in diabetic patients.

The combination of these materials with modified PCL nanofibers [54,55,56] can further improve the healing process, going beyond the state-of-the-art.

## Figures and Tables

**Figure 1 pharmaceutics-14-00724-f001:**
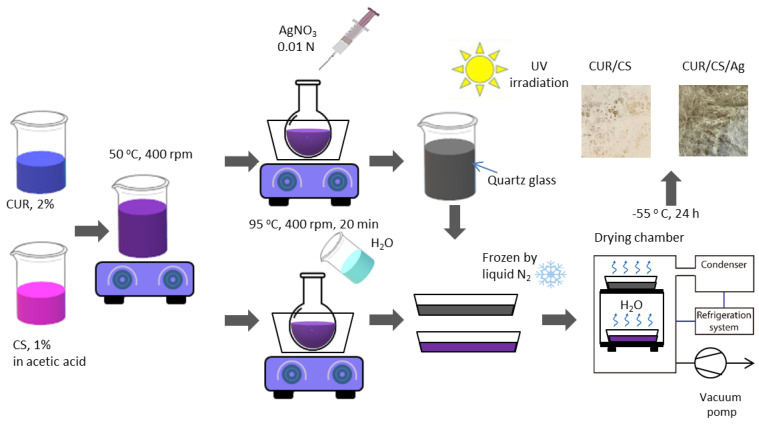
Schematic of curdlan/chitosan and curdlan/chitosan/Ag foam synthesis.

**Figure 2 pharmaceutics-14-00724-f002:**
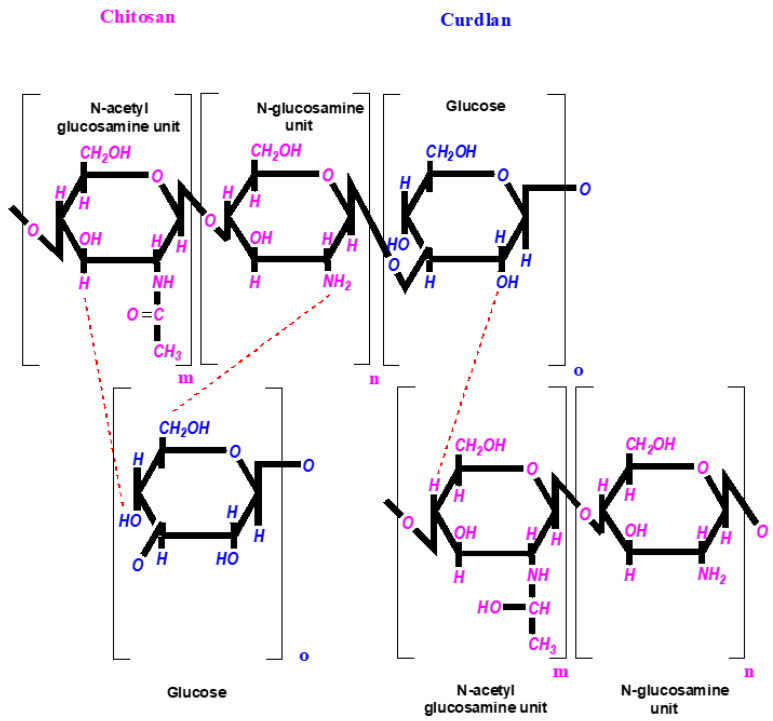
Schematics of the possible chemical interactions between curdlan and chitosan.

**Figure 3 pharmaceutics-14-00724-f003:**
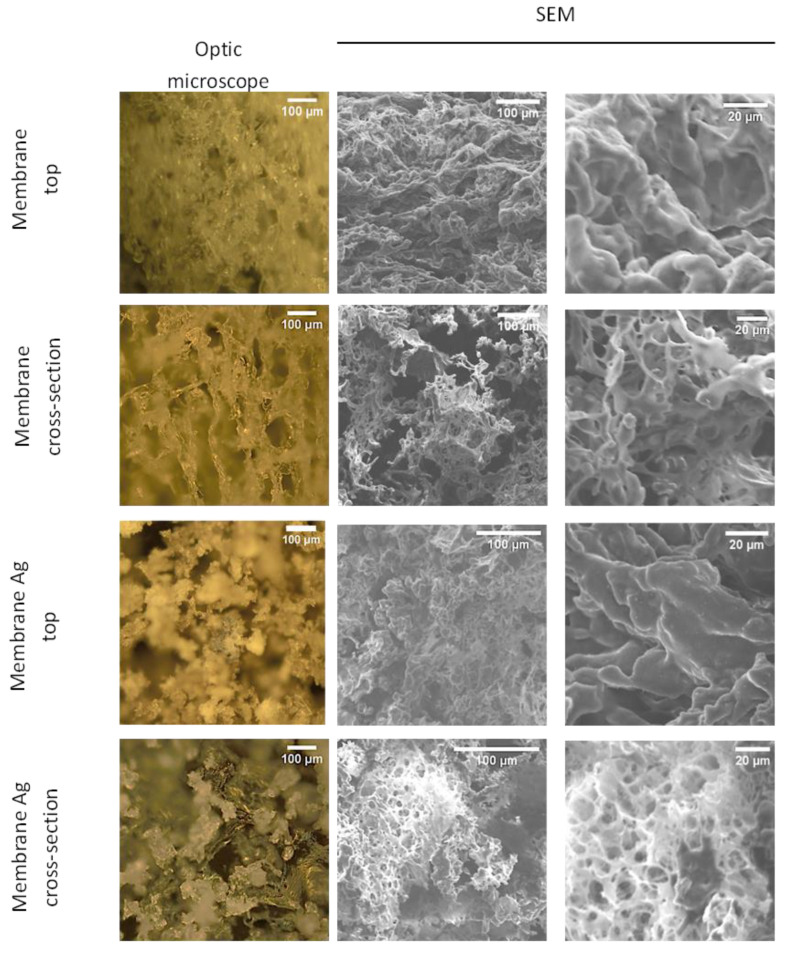
Optical and SEM images of CUR/CS and CUR/CS/Ag samples.

**Figure 4 pharmaceutics-14-00724-f004:**
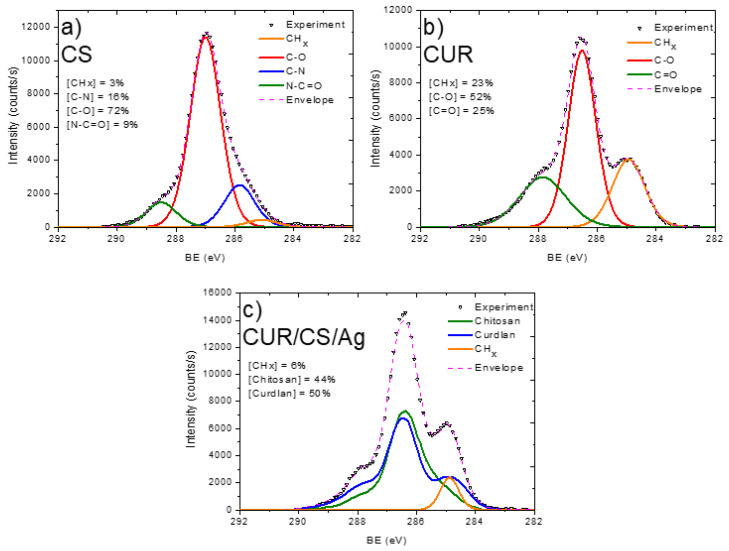
XPS C1s curve fitting of CS (**a**), CUR (**b**), and CS-CUR-Ag (**c**).

**Figure 5 pharmaceutics-14-00724-f005:**
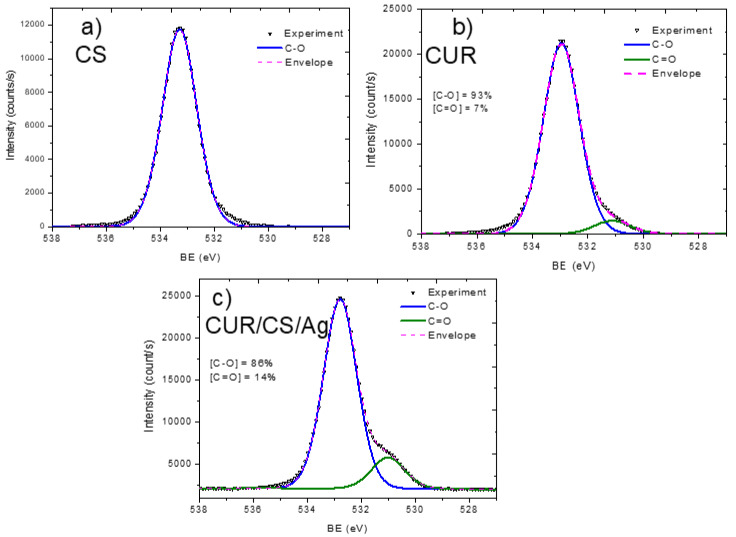
XPS O1s curve fitting of CS (**a**), CUR (**b**), and CS-CUR-Ag (**c**) samples.

**Figure 6 pharmaceutics-14-00724-f006:**
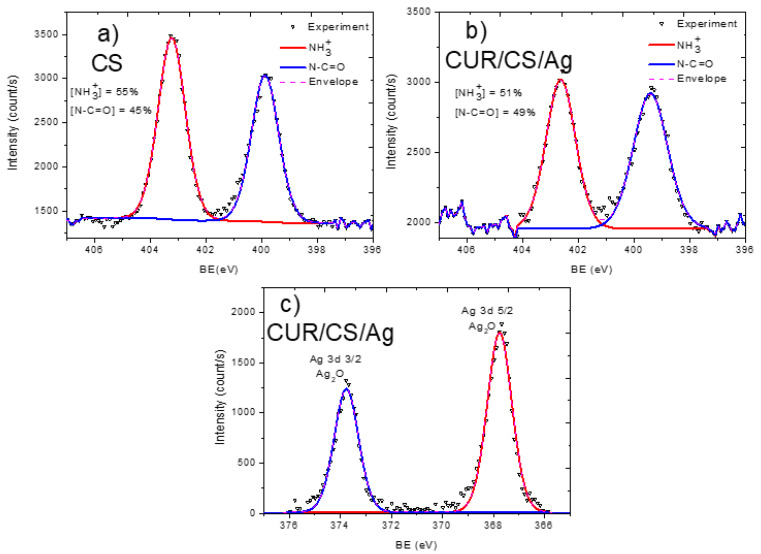
XPS curve fitting of N1s signal of CS (**a**) and CS-CUR-Ag (**b**) samples, and Ag 3D spectrum of CS-CUR-Ag (**c**).

**Figure 7 pharmaceutics-14-00724-f007:**
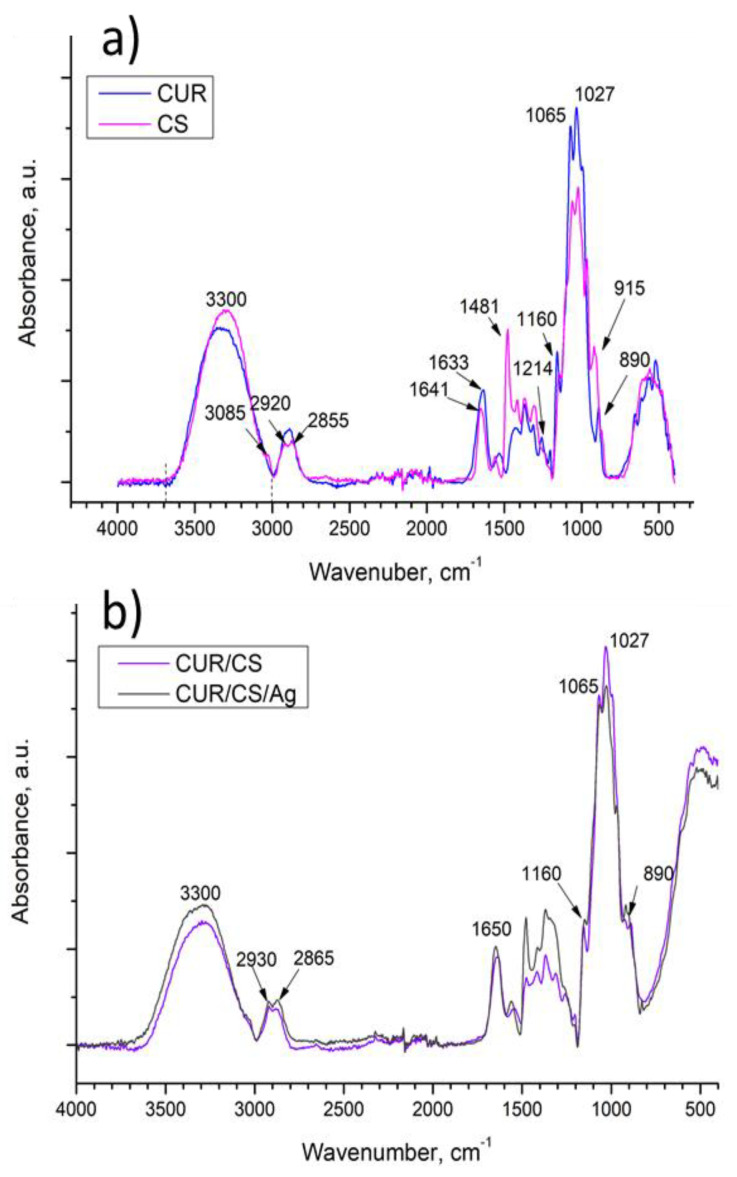
FTIR spectra of curdlan-chitosan (**a**), and CUR/CS and CUR/CS/Ag (**b**) biomaterials.

**Figure 8 pharmaceutics-14-00724-f008:**
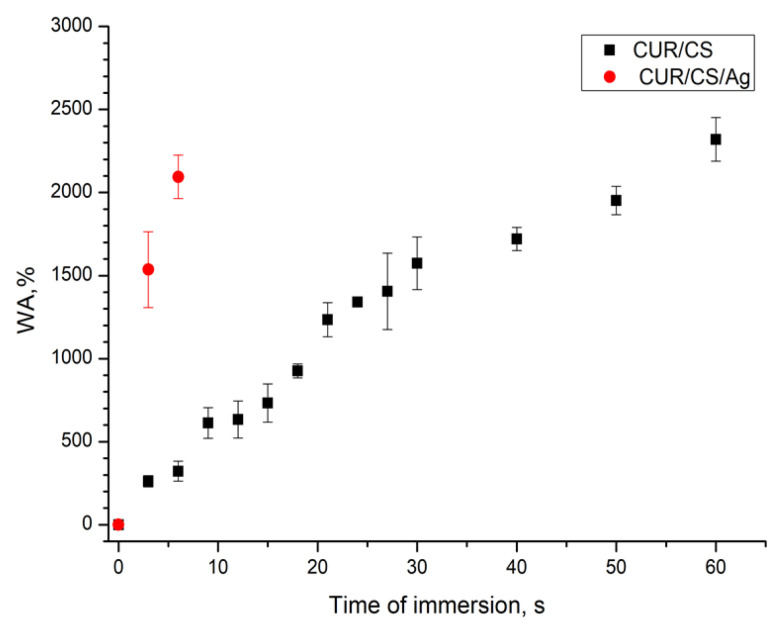
PBS absorption capacity of the CUR/CS and CUR/CS/Ag biomaterials.

**Figure 9 pharmaceutics-14-00724-f009:**
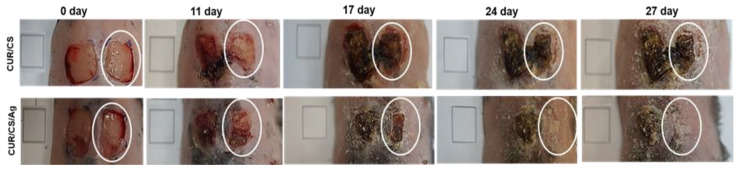
The representative photographs showing the healing dynamic of full-thickness skin with (**right wound**) or without (**left wound**) treatment with curdlan/chitosan and curdlan/chitosan/Ag foams, for each of the groups of animals, n = 3. The ovals show the foams of the treated wounds. Wounds were covered with biomaterials for 10 days, after which healing occurred without the influence of biomaterials.

**Figure 10 pharmaceutics-14-00724-f010:**
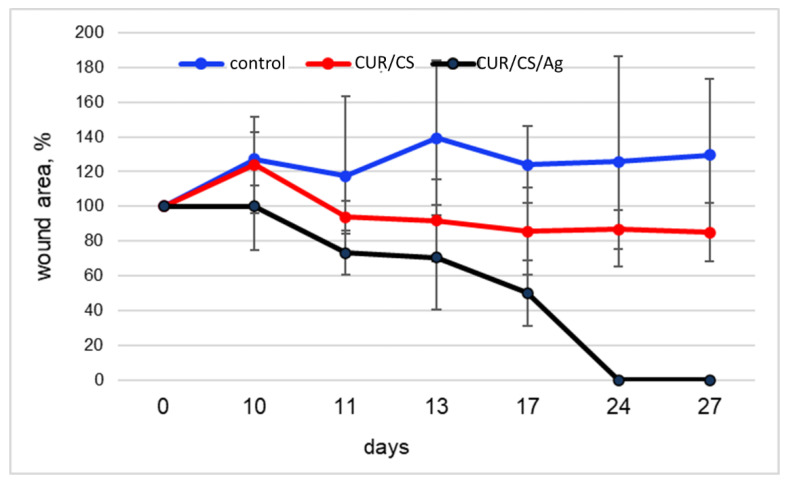
The influence of membranes (red—CUR/CS; black—CUR/CS/Ag; blue—untreated control) on the dynamics of the wound defect closure. The graph shows the dynamics of the closure of a full-thickness skin defect in mice with genetically determined type 2 diabetes mellitus. Each animal (n = 3 in each group in total) received a control wound (without therapy) and an experimental wound (treatment with test materials). The percentage of closure was calculated based on the initial size of the wounds in order to assess the actual dynamics of healing. The graph shows the increase in the sparing of the control wounds for all observation times, as well as wounds with CUR/CS therapy on day 10 (removal of the dressing).

**Figure 11 pharmaceutics-14-00724-f011:**
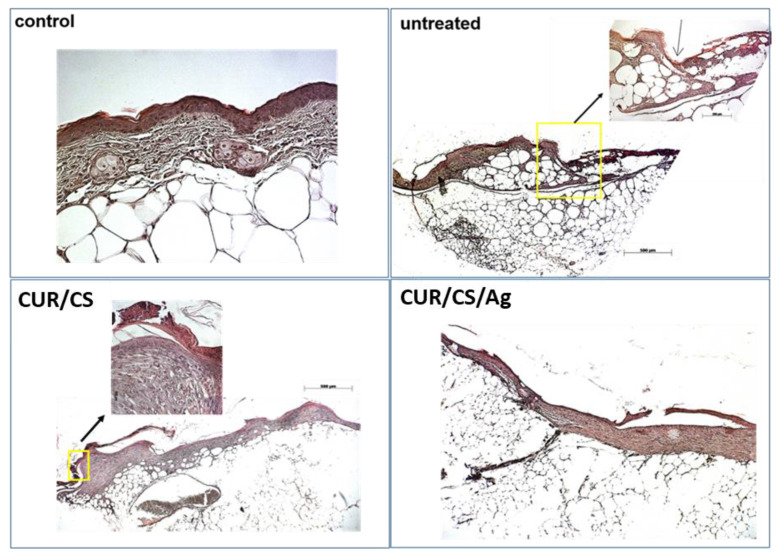
Histology analysis of a full-thickness cutaneous defect after treatment with membranes. Yellow squares enclose the edge of the wound. Red arrows indicate infiltrating inflammatory cells (polymorphonuclear lymphocytes; number of animals in each group of dressings, n = 3).

**Table 1 pharmaceutics-14-00724-t001:** Atomic percentage (%) from energy dispersive X-ray (E.D.X.) element mapping.

Samples	Atomic Percentage (%)				
	C	O	N	Ag	Pt
CUR/CS	53.0	41.0	5.8	-	0.2
CUR/CS-Ag	52.6	40.2	6.6	0.4	0.2

**Table 2 pharmaceutics-14-00724-t002:** Atomic percentages (%) obtained from the XPS surface analyses.

Atomic Percentage (%)	Samples			
	CS	CUR	CUR/CS	CUR/CS/Ag
C	68.6	62.8	60.6	63.7
O	22.6	37.2	31.2	32.1
N	8.8	0.0	8.2	3.8
Ag	0.0	0.0	0.0	0.4

## Data Availability

Data are available from corresponding author upon reasonable request.

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
