# Peer review of "Ag-Contained Superabsorbent Curdlan–Chitosan Foams for Healing Wounds in a Type-2 Diabetic Mice Model"

_pharmaceutics, 2022, doi:10.3390/pharmaceutics14040724_

Round 1

Reviewer 1 Report

This manuscript focused on the synthesis and characterization of purdlan-chitosan foams containing Ag nanoparticles for the purpose of skin repair of diabetic mice. The results support the conclusions well and the manuscript is almost well organized. However, after publication, three comments should be sloved.

  1. The the potentially trace remained AgNO3 has negative effects on health and practical effects?
  2. How to ensure that silver nanoparticles will not fall off during use? If it falls off, will it have a negative impact on the effect?
  3. The background of this manuscript should be enhanced. The authors should give us strong reasons that they use Ag nanoparticles but not other inorganic materials, since Ag possesses high cost. Some reference is suggested, Composites part B, 226, 109335.

Author Response

First of all, we highly appreciate essential and useful comments from rspected reviewers and deeply thank them for valuable suggestions on the structure and content of our manuscript. We have modified the manuscript accordingly, indicated all changes in the text made in response to the reviewers. The detailed corrections are listed below in a point-by-point style.

Reviewer: 1

Comment 1: This manuscript focused on the synthesis and characterization of purdlan-chitosan foams containing Ag nanoparticles for the purpose of skin repair of diabetic mice. The results support the conclusions well and the manuscript is almost well organized. However, after publication, three comments should be sloved.

  1. The potentially trace remained AgNO3 has negative effects on health and practical effects?

Author response

It should be noted that the dressing CUR/CS/Ag doesn’t contain AgNO3, since AgNO3 was reduced to Ag NPs by UV-light after the polymerization process (NO3 part was solved in the water and evaporated during the lyophilization). Guo et al. [1] compared the toxic effect of AgNO3 and Ag NPs in vitro and in vivo. It was shown that AgNO3 is significant more toxic than Ag NPs, that could be explained by different mechanisms of action: AgNO3-mediated cell death in the cell medium was inducement of impairment of cell membrane permeability

to ions K+ and Na+ while Ag NPs can be taken up by endothelial cells and induced concentration-dependent intracellular ROS elevation, which could be rescued by antioxidant N-acetylcysteine (NAC).  The final concentration of Ag NPs in the composition CUR/CS/Ag was ultralow (≈0.4% wt.) that was determined by EDS and XPS analysis. Ag NPs incorporated in the structure of CUR/CS/Ag biomaterial and can be released only in the case of high exudate wounds with the parallel process of dissolution of material.  Even in that case it doesn’t mean that all Ag NPs would interact with skin layer or penetrate to the bloodstream. But in this way  Ag NPs probably will be coated by polymer layer since ionic bonds are stronger than hydrogen bonds. Kim et al. [2]conducted dermal toxicity/irritation tests on rats, rabbits and guinea pigs using  сitrate-coated 10nm AgNP. Ten rats were exposed for 24 hours to up to 2000mg/kg bw and then observed for 15 days; no toxicity was observed. Similarly, no skin reaction was seen in three rabbits subjected to the same form of AgNP. In a skin sensitisation test using 20 guinea pigs, a single animal showed some erythema, suggesting that the tested AgNP could be classified as a weak skin sensitiser. According to report WHO (Microsoft Word - Silver water disinfection and toxicity_2014V2.docx (who.int)) silver may have toxic effects via ingestion but in high concentrations (at least more than 100 mg).

Finally, it should be noted that NO3 peak is absent in the N1s spectrum and thus we may consider the there is no noticeable nitrate contaminations.

Comment 2: How to ensure that silver nanoparticles will not fall off during use? If it falls off, will it have a negative impact on the effect?

Author response

The silver nanoparticles incorporated in the structure of obtained CUR/CS/Ag foams. AgNO3 solution was added during the polymerization process thus Ag NPs coordinated with functional groups of polymers and equally distributed throughout the CUR/CS/Ag material. In this way Ag NPs can be released only in the case of high exudate wounds with the parallel process of dissolution of sample. Predictably that in this case Ag NPs can penetrate in the skin or bloodstream. However according to the reviews [3,4] there have been very few confirmed cases of contact dermatitis secondary to silver-containing wound products like silver sulfadiazine and skin markers that contain silver nitrate.

Comment 3: The background of this manuscript should be enhanced. The authors should give us strong reasons that they use Ag nanoparticles but not other inorganic materials, since Ag possesses high cost. Some reference is suggested, Composites part B, 226, 109335.

Author response

The concentration of Ag NPs in CUR/CS/Ag is ultralow (0.4 wt%) thus 250 g CUR/CS/Ag will contain just 1 g Ag NPs so it corresponds to 2,5 m2 of CUR/CS/Ag material that makes using Ag NPs in this material economically viable. The recommended reference is added (ref43).

The next paragraph was added

“Although silver-containing wound products like 1% cream silver sulfadiazine or Ag-contained dressings (AQUACEL® Ag Extra™ dressings, ConvaTec;  Silver Alginate, Areza Medical; ATRAUMAN AG, Hartmann; etc) are widely used in diabetic wound healing application and demonstrate a good results, the toxicity of Ag ions and Ag NPs is still hot topic in the scientific field. In some articles [4–6] reported about mechanisms of toxicity Ag NPs included involved oxidative stress, genotoxicity, activation of lysosomal activity, disruption of actin cytoskeleton and stimulation of phagocytosis, increase of MXR transport activity and inhibition of Na-K-ATPase. However the main contribution in toxicity of Ag NPs was determined  as takeover AgNPs by endothelial cells and induced concentration-dependent intracellular ROS elevation. It was shown that covering Ag NPs by polymer or biomolecules (polyvinylpyrrolidone[7,8], citrate[7,9],tyrosine[6], PEG[8] etc) decrease the cytotoxicity. In this research Ag NPs incorporated in the structure of CUR/CS/Ag material, where antibacterial polymers (curdlan, chitosan) decrease required concentration of Ag NPs and most probably decrease the cytoxicity of Ag NPs due to binding organic molecules to the metal surface through nucleophilic functional groups. However, alternative antibacterial nanoparticles are also being actively developed for use in dressings, such as ZnO[10,11], Cu[12], Fe3O4 [13]and etc. In our future research we are also planning to test the alternative nanoparticles.”

  1. Guo, H.; Zhang, J.; Boudreau, M.; Meng, J.; Yin, J.; Liu, J.; Xu, H. Intravenous administration of silver nanoparticles causes organ toxicity through intracellular ROS-related loss of inter- endothelial junction. Part. Fibre Toxicol. 2016, 1–13, doi:10.1186/s12989-016-0133-9.
  2. Kim, J.S.; Song, K.S.; Sung, J.H.; Ryu, H.R.; Gil, B.; Cho, H.S.; Lee, J.K.; Yu, I.J. Genotoxicity , acute oral and dermal toxicity , eye and dermal irritation and corrosion and skin sensitisation evaluation of silver nanoparticles. Nanotoxicology 2013, 7, 953–960, doi:10.3109/17435390.2012.676099.
  3. Response, I.; Matteis, V. De Exposure to Inorganic Nanoparticles: Routes of Entry, Immune Response, Biodistribution and In Vitro/In Vivo Toxicity Evaluation. Toxics 2017, 5, 1–21, doi:10.3390/toxics5040029.
  4. Ferdous, Z.; Nemmar, A. Health Impact of Silver Nanoparticles : A Review of the Biodistribution and Toxicity Following Various Routes of Exposure. Int. J. Mol. Sci. 2020, 21, 2375, doi:doi:10.3390/ijms21072375.
  5. Katsumiti, A.; Gilliland, D.; Arostegui, I.; Cajaraville, M.P. Mechanisms of Toxicity of Ag Nanoparticles in Comparison to Bulk and Ionic Ag on Mussel Hemocytes and Gill Cells. 2015, 1–30, doi:10.1371/journal.pone.0129039.
  6. Lekamge, S.; Miranda, A.F.; Abraham, A.; Li, V.; Shukla, R.; Bansal, V.; Nugegoda, D. The Toxicity of Silver Nanoparticles ( AgNPs ) to Three Freshwater Invertebrates With Different Life Strategies : Hydra vulgaris , Daphnia carinata , and Paratya australiensis. Front. Environ. Sci. 2018, 6, 152–165, doi:10.3389/fenvs.2018.00152.
  7. Moon, T. Comparison of toxicity of uncoated and coated silver nanoparticles. J. Phys. Conf. Ser. 2013, 49, 012025, doi:10.1088/1742-6596/429/1/012025.
  8. Xiu, Z.; Zhang, Q.; Puppala, H.L.; Colvin, V.L.; Alvarez, P.J.J. Negligible Particle-Specific Antibacterial Activity of Silver Nanoparticles. Nano Lett. 2012, 12, 4271–4275.
  9. Mitra, C.; Gummadidala, P.M.; Afshinnia, K.; Corrin, R.; Baalousha, M.A.; Lead, J.R.; Chanda, A. Citrate-coated silver nanoparticles growth- independently inhibit aflatoxin synthesis in Aspergillus parasiticus. Environ. Sci. Technol 2017, 51, 8085–8093.
  10. Melnikova, N.; Knyazev, A.; Nikolskiy, V.; Peretyagin, P.; Belyaeva, K.; Nazarova, N.; Liyaskina, E.; Malygina, D.; Revin, V. Wound Healing Composite Materials of Bacterial Cellulose and Zinc Oxide Nanoparticles with Immobilized Betulin Diphosphate. Nanomaterials 2021, 11, 713.
  11. Alexandra, C.; Rayyif, S.M.I.; Mohammed, H.B.; Curut, C.; Chifiriuc, M.C.; Mih, G.; Holban, A.M. ZnO Nanoparticles-Modified Dressings to Inhibit Wound Pathogens. Materials (Basel). 2021, 14, 3084.
  12. Webster, T.J. Antimicrobial Double-Layer Wound Dressing Based on Chitosan / Polyvinyl Alcohol / Copper : In vitro and in vivo Assessment. Int. J. Nanomedicine 2021, 16, 223–235.
  13. Lou, Z.; Han, X.; Liu, J.; Ma, Q.; Yan, H.; Yuan, C.; Yang, L.; Han, H.; Weng, F.; Li, Y. Nano-Fe3O4 / bamboo bundles / phenolic resin oriented recombination ternary composite with enhanced multiple functions. Compos. Part B 2021, 226, 109335, doi:10.1016/j.compositesb.2021.109335.

Reviewer 2 Report

  1. In this paper, the discussion section is missing. 
  2. A lot of typos. Please check carefully all manuscripts.
  3. Why wounds weren't protected by sutures ?
  4. In which mechanism do wounds heal? By contraction or epithelization?
  5. Fig. 10 should be presented more clearly. Maybe another type of graph should be applied. 
  6. The histological results are very briefly described. Please investigated what type of cells infiltrated wounds. 
  7. The Author told that they do not observe any inflammatory reaction. On what grounds do you make such a claim?
  8. Do you perform any PCR/WB or ICH tests to confirm your thesis?
  9.  Do you perform any microbiology tests to confirm their antibacterial properties? I recommended doing a basic microbiological test.
  10. The molecular study should be performed to confirm the role of obtained dressing e.g. mTOR/AKT pathway. 

Author Response

We highly appreciate essential and useful comments from respected reviewers and deeply thank them for valuable suggestions on the structure and content of our manuscript. We have modified the manuscript accordingly, indicated all changes in the text made in response to the reviewers. The detailed corrections are listed below in a point-by-point style. For more convenience you may use the attached doc file.

Reviewer:3

Comment 1: In this paper, the discussion section is missing.

Author response

The discussion section was added to the article

Although silver-containing wound products like 1% cream silver sulfadiazine or Ag-contained dressings (AQUACEL® Ag Extra™ dressings, ConvaTec; Silver Alginate, Areza Medical; ATRAUMAN AG, Hartmann; etc) are widely used in diabetic wound healing application and demonstrate a good results, the toxicity of Ag ions and Ag NPs is still hot topic in the scientific field. In some articles [4–6] reported about mechanisms of toxicity Ag NPs included involved oxidative stress, genotoxicity, activation of lysosomal activity, disruption of actin cytoskeleton and stimulation of phagocytosis, increase of MXR transport activity and inhibition of Na-K-ATPase. However the main contribution in toxicity of Ag NPs was determined as takeover Ag NPs by endothelial cells and induced concentration-dependent intracellular ROS elevation. [1] It was shown that covering Ag NPs by polymer or biomolecules (polyvinylpyrrolidone[7,8], citrate[7,9],tyrosine[6], PEG[8] etc) decrease the cytotoxicity. In this research Ag NPs incorporated in the structure of CUR/CS/Ag material, where antibacterial polymers (curdlan, chitosan) decrease required concentration of Ag NPs and most probably decrease the cytoxicity of Ag NPs due to binding organic molecules to the metal surface through nucleophilic functional groups. In other side the rapid release of Ag ions can promote the activation of specific immunocytes. In particular, several studies have demonstrated that AgNPs can directly activate the innate immune response, especially macrophages [14,15].

In Table X summarized some recent research aimed at development new dressings for wound healing. It should be noted that the regeneration rate depends on the type (burn or wound) and size of the skin disruption and also inflammation processes or diabetic type of wound. In addition to silver nanoparticles, the dressings also include nanoparticles ZnO[10,11], Cu[12, 20], Fe3O4 [13], Ca[16].

In the clinic research [17] reported about influence copper oxide microparticles (CuO MPs) on healing diabetic foot ulcer (13 patients). The research is to soak standard of care dressings by CuO MPs and then evaluate the differences of closure rate. It was shown that CuO MPs stimulate the healing of non-infected stagnated wounds in diabetic patients.

In another clinical article dressing based on chitosan hydrogel loaded by Ag NPs and calendula extract was tested to heal chronic wounds (2 patients). [18] Using this dressings (change every 7d due to 2 weeks) decreased considerably the pain and inflammation, until the symptoms were eliminated. At the end of the 15 days of treatment, it was observed that these had adapted to the size of the wound and, remained completely adhered, so it was decided to leave them for a period of 15 days. After this period, it was observed that the wounds had healed.

Recent research are also demonstrate the effectivity of using bioactive molecules such as growth factors[19,20], enzyme[21] or receptors[22] in the composition of dressings to enhance therapy of chronic wounds.

Table X Recent studies of development dressing for wound healing

Dressings

Size of wound

Effect in vivo

Animal

Ref

1

Structure: nanofibers had a diameter between 200 to 300 nm, size of NPs 50-100nm

Composition: Chitosan/Polyvinyl Alcohol/Copper NPs

Wound

1.5cm×1.5cm Area:225 mm2

The wound closure rate of the negative control group was 18.46%, 59.89%, 62.42% , and 88.07% and the wound closure rate of the positive control group was 25.33%, 72.85%, 95.32%, and 97.90% for 3, 7, 11 and 15 days, respectively.

Rat

[12]

2

Structure: film and gel functionalized by NPs(11.5–18.71 nm)

Composition: bacterial cellulose/ betulin diphosphate/ ZnO NPs

Burn

Area:1400 ± 50 mm2

Depth 3–5 mm

On day 21, the wound area treated with BC-ZnO NPs-BDP films was reduced by 34.3%, while when treated with ZnO NPs-BDP oleogel, a large decrease up to 40.6% was observed. In untreated control the closure rate was just 19,2%

Rat

[10]

3

Structure: electrospun fibers (648.1 ± 72.2 nm) with NPs

Composition: PLA + Ca NPs

12 mm square skin wounds Area 452 mm2

80% contraction in wound area vs 62% in untreated control on 8 day

Diabetic mice

[16]

4

Structure: hydrogel with NPs (size 99.1 ± 2.3 nm)

Composition: chitosan/PEG/Ag NPs

20 mm square skin wounds

Area 1256 mm2

47.7 ± 1.8% contraction in wound area was recorded with the AgNPs impregnated chitosan-PEG hydrogel group as compared to 12.6 ± 1.3% in negative control

Diabetic

rabbit

[23]

5

Standard of care dressings impregnated with copper oxide microparticles (COD)

9.26 ± 6.9 cm2 (range of 1.35–23.6 cm2)

Following 1 month of copper improved treatment, there was a clear reduction in the mean wound area (53.2%; p = 0.003), an increase in granulation tissue (43.37; p < 0.001), and a reduction in fibrins (47.8%; p = 0.002). In control group wound closure was less than 20%

Clinic diabetic foot ucler

[17]

6

Structure: hydrogel

Composition: gelatin/hyaluronic acid/thrombomodulin

8-mm diameter round-shaped wound

Area:201 mm2

On 10 day wound closure was 80% for hydrogel with thrombomodulin vs 40% untreated control

Mice

[22]

7

Structure: electrospun fibers (90-120 nm)

Composition:Enteromorpha polysaccharide and polyvinyl alcohol (PVA)

10-mm diameter round-shaped wound

Area: 314 mm2

On day 9, the wound contraction rate of the PVA/EPP1 group reached nearly 72% vs 54% of control group

Diabetic

mice

[24]

8

Structure: electrospun fibers (110 ± 74 nm)

Composition: hydroxypropyl methylcellulose (HPMC) / polyethylene oxide (PEO)/ Beta-glucan

1 cm×1 cm

Area: 100 mm2

βG-nanofibers 95% healing versus 40% healing of control in 24 days

Diabetic mice

[25].

9

Structure: hydrogel Composition: chitosan, heparin and poly (γ-glutamic acid) and loaded with superoxide dismutase

10-mm diameter round-shaped wound

Area: 314 mm2

after 21 days closure rate is 92.0% ± 3.7% compared with the control group (85.4% ± 2.4%)

Diabetic mice

[21]

10

Structure: hydrogel Composition: glycol chitosan, loaded by growth factors (VEGF and PDGF-BB)

5-mm diameter round-shaped wound

Area: 157 mm2

On 3 day hydrogel dressing demonstrate 60%closure rate vs less than 5% of Duoderm dressing

Diabetic mice

[20]

The present study examined the ability of Ag-contained CUR/CS foams to accelerate the wound healing in diabetic mice. It has been shown that Ag NPs in CUR/CS foams significantly affect the PBS absorbance capacity (the swelling rate increases by ~10 times) and accelerate the healing process of diabetic wound.

The wounds treated with CUR/CS/Ag showed complete healing after 24 days, which was accompanied by the formation of new vessels of different diameters and significant re-epithelialization. At the same time, CUR/CS-induced healing was only 20%, and the control showed an increase in the area of wound by 20+%, which is associated with inflammatory processes. It is possible that such a difference in the effect is associated with different material dissolution rates. The developed CUR/CS/Ag foams could meet the needs for clinical practice, and may have future medical applications for wound care especially in diabetic patients.

  1. A lot of typos. Please check carefully all manuscripts.

We do apologize for the mistakes and typos in our previous version. The manuscript was significantly revised

  1. Why wounds weren't protected by sutures ?

These materials are being developed for the treatment of long-term healing ulcers in patients with diabetes mellitus. Ulcers are characterized by chronic inflammation, impaired dynamics of the wound process, and lack of regeneration. Therapy usually consists of removing necrotic tissue, followed by the application of various therapeutic dressings. Suturing is used only for amputations. Therefore, we used full-thickness wounds without suturing in our work

  1. In which mechanism do wounds heal? By contraction or epithelization?

Wound healing in mice proceeds in two ways : 1) an epithelialization, through the formation of granulation tissue, which is clearly visible on histological sections, 2) a contraction, since mice have an additional subcutaneous muscle layer - panniculus carnosus.

  1. Fig. 10 should be presented more clearly. Maybe another type of graph should be applied.

We thank the reviewer for this valuable comment that we certainly took into account. The explanatory information was added in the caption to the figure:

“This graph shows the dynamics of the closure of a full-thickness skin defect in mice with genetically determined type 2 diabetes mellitus. Each animal (n=3 in each group in total) received a control wound (without therapy) and an experimental wound (treatment with test materials). The percentage of closure was calculated based on the initial size of the wounds in order to assess the actual dynamics of healing. The graph shows the increase in the sparing of the control wounds for all observation time, as well as wounds with CUR/CS therapy on day 10 (removal of the dressing).”

  1. The histological results are very briefly described. Please investigated what type of cells infiltrated wounds.

The histological analysis was done to follow the course of the wound process using the developed materials according to the standard methodology. The main focus was on the presence of inflammation (lymphocyte infiltration), epithelialization and angiogenesis [ Estevão, L.; Cassini-Vieira, P.; Leite, A.G.; Bulhões, A.; Barcelos, L. da; Evêncio-Neto, J. Morphological Evaluation of Wound Healing Events in the Excisional Wound Healing Model in Rats. Bio-Protocol 20199, 1–12]. The images were added to the article, showing in more detail the infiltration by inflammatory cells (polymorphonuclear leukocytes and mononuclears in the control, which are easily identified by the shape of their nuclei (horseshoe-shaped, bean-shaped, etc.), intensely stained with hematoxylin).

  1. The Author told that they do not observe any inflammatory reaction. On what grounds do you make such a claim?

Thank you for your comment and question. The inflammatory response was assessed by the presence of lymphoid cells in the wound. Photographs were added to the article, showing in more detail the infiltration by inflammatory cells (polymorphonuclear leukocytes and mononuclears in the control, which are easily identified by the shape of their nuclei (horseshoe-shaped, bean-shaped, etc.), stained with hematoxylin).

  1. Do you perform any PCR/WB or ICH tests to confirm your thesis?

We based our conclusions on the basis of a histological analysis performed according to a standard technique [ Estevão, L.; Cassini-Vieira, P.; Leite, A.G.; Bulhões, A.; Barcelos, L. da; Evêncio-Neto, J. Morphological Evaluation of Wound Healing Events in the Excisional Wound Healing Model in Rats. Bio-Protocol 20199, 1–12]. This methodology is widely employed and generally accepted. Nevertheless, we do appreciate this suggestion and we will consider to expand the techniques in our future research.

  1. Do you perform any microbiology tests to confirm their antibacterial properties? I recommended doing a basic microbiological test.

WE do appreciate this the recommendation and we plan to further study these materials on models of infected wounds, in which we plan to investigate in detail the antibacterial effect of the obtained materials. In this work, no bacterial colonization of the wound surface was found. However, we assume that these materials have a combined antibacterial effect. In addition to the direct antibacterial effect of AgNPs, it stimulate of the lymphoid cells and macrophages migration, which are the first line of defense against bacterial invasion [Troy, E.; Tilbury, M.A.; Power, A.M.; Wall, J.G. Nature-based biomaterials and their application in biomedicine. Polymers (Basel). 202113, 1–37.

Tang, J.; Zhen, H.; Wang, N.; Yan, Q.; Jing, H.; Jiang, Z. Curdlan oligosaccharides having higher immunostimulatory activity than curdlan in mice treated with cyclophosphamide. Carbohydr. Polym. 2019207, 131–142]. Also, CUR/CS/Ag stimulate faster formation of a scab, which has a protective effect on the wound surface

  1. The molecular study should be performed to confirm the role of obtained dressing e.g. mTOR/AKT pathway.

Indeed, the mTOR/AKT pathway is implicated in the development of diabetes mellitus, especially in cardiovascular and renal complications. It is known that chitosan and curdlan affect the proliferation of fibroblasts and keratinocytes indirectly, through its association with growth factors (see refs: Tang, J.; Zhen, H.; Wang, N.; Yan, Q.; Jing, H.; Jiang, Z. Curdlan oligosaccharides having higher immunostimulatory activity than curdlan in mice treated with cyclophosphamide. Carbohydr. Polym. 2019207, 131–142 AND Howling, G.I.; Dettmar, P.W.; Goddard, P.A.; Hampson, F.C.; Dornish, M.; Wood, E.J. The effect of chitin and chitosan on the proliferation of human skin fibroblasts and keratinocytes in vitro. Biomaterials 200122, 2959–2966.), and therefore do not directly affect this pathway. Accordingly, we do not consider it appropriate to determine this path. In this work, we assume that the positive effect of the materials comes through the activation of the migration of lymphoid cells and macrophages in the initial period of healing through IL-8 and the complement component C5a, antibacterial property of AgNPs and the acceleration of wound cleansing, respectively, as well as influence to fibroblasts proliferation at regeneration (proliferation) stage of wound healing process

Reviewer 3 Report

Review comments:

This is a well-organized and well-illustrated paper, has an important clinical message, and should be of great interest to the readers. This paper evaluated the potential of Ag-contained superabsorbent curdlan-chitosan foams for wound dressing applications in type-2 diabetic mice. This research opens new possibilities for the development of new antimicrobial formulations for wound dressing in diabetic patients and other clinical applications. This manuscript deserves publication after addressing some minor issues cited below.

  1. I suggest the authors to slightly revise the manuscript title by including the terms “in a type-2 diabetic mice model” in the title of the paper though the proposed formulation have a wide range of antimicrobial applications in various wound dressings. This can increase the visibility of paper and increase the citations for this manuscript, as very few reports study the efficacy of nano formulations in vivo.
  2. The authors need to mention animal numbers “n” value in figure 9,10 and 11.
  3. Can the combination of curdlan and chitosan potentiate the antimicrobial properties or enhance the stability of the formulation? If yes, please mention briefly in the introduction.
  4. Can the proposed formulation cause any skin irritation or inflammation on the animal skin? There are a few reports of silver induced inflammation in wound dressing formulations. Is the combination of chitosan and curdlan help in alleviating these side effects? If yes discuss this in the discussion section.

Author Response

First of all, we highly appreciate essential and useful comments from respected reviewers and deeply thank them for valuable suggestions on the structure and content of our manuscript. We have modified the manuscript accordingly, indicated all changes in the text made in response to the reviewers. The detailed corrections are listed below in a point-by-point style.

Reviewer: 2

This is a well-organized and well-illustrated paper, has an important clinical message, and should be of great interest to the readers. This paper evaluated the potential of Ag-contained superabsorbent curdlan-chitosan foams for wound dressing applications in type-2 diabetic mice. This research opens new possibilities for the development of new antimicrobial formulations for wound dressing in diabetic patients and other clinical applications. This manuscript deserves publication after addressing some minor issues cited below.

Comment 1:I suggest the authors to slightly revise the manuscript title by including the terms “in a type-2 diabetic mice model” in the title of the paper though the proposed formulation have a wide range of antimicrobial applications in various wound dressings. This can increase the visibility of paper and increase the citations for this manuscript, as very few reports study the efficacy of nano formulations in vivo.

Author response

The title of article was changed according to the recommendation: “Ag-contained superabsorbent curdlan-chitosan foams for healing wounds in a type-2 diabetic mice model

Comment 2:

The authors need to mention animal numbers “n” value in figure 9,10 and 11.

Author response

The animal numbers were added in figure 9,10 and 11 and also in description of figures.

Comment 3: Can the combination of curdlan and chitosan potentiate the antimicrobial properties or enhance the stability of the formulation? If yes, please mention briefly in the introduction.

Author response

The next paragraph was added

““Although silver-containing wound products like 1% cream silver sulfadiazine or Ag-contained dressings (AQUACEL® Ag Extra™ dressings, ConvaTec; Silver Alginate, Areza Medical; ATRAUMAN AG, Hartmann; etc) are widely used in diabetic wound healing application and demonstrate a good results, the toxicity of Ag ions and Ag NPs is still hot topic in the scientific field. In some articles [1–3] reported about mechanisms of toxicity Ag NPs included involved oxidative stress, genotoxicity, activation of lysosomal activity, disruption of actin cytoskeleton and stimulation of phagocytosis, increase of MXR transport activity and inhibition of Na-K-ATPase. However the main contribution in toxicity of Ag NPs was determined as takeover AgNPs by endothelial cells and induced concentration-dependent intracellular ROS elevation. It was shown that covering Ag NPs by polymer or biomolecules (polyvinylpyrrolidone[4,5], citrate[4,6],tyrosine[3], PEG[5] etc) decrease the cytotoxicity. In this research Ag NPs incorporated in the structure of CUR/CS/Ag material, where antibacterial polymers (curdlan, chitosan) decrease the required concentration of Ag NPs and most probably decrease the cytoxicity of Ag NPs due to binding organic molecules to the metal surface through nucleophilic functional groups. However, alternative antibacterial nanoparticles are also being actively developed for use in dressings, such as ZnO[7,8], Cu[9], Fe3O4 [10]and etc. In our future research we are also planning to test the alternative nanoparticles.”

Comment 4: Can the proposed formulation cause any skin irritation or inflammation on the animal skin? There are a few reports of silver induced inflammation in wound dressing formulations. Is the combination of chitosan and curdlan help in alleviating these side effects? If yes discuss this in the discussion section.

Author response

The sample CUR/CS/Ag didn’t demonstrate skin irritation or inflammation on the animal skin in our in vivo tests including histological analysis, that could be explained by ultralow concentration of Ag NPs in the composition CUR/CS/Ag (less than 0.4% wt.). Ag NPs incorporated in the structure of CUR/CS/Ag biomaterial and can be released only in the case of high exudate wounds with the parallel process of dissolution of material. Even in that case it doesn’t mean that all Ag NPs would interact with skin layer or penetrate to the bloodstream. But in this way metallic surface of Ag NPs probably is bonded with nucleophilic functional groups of polymers. Kim et al. [11]conducted dermal toxicity/irritation tests on rats, rabbits and guinea pigs using сitrate-coated 10nm AgNP. Ten rats were exposed for 24 hours to up to 2000mg/kg bw and then observed for 15 days; no toxicity was observed. Similarly, no skin reaction was seen in three rabbits subjected to the same form of AgNP. In a skin sensitisation test using 20 guinea pigs, a single animal showed some erythema, suggesting that the tested AgNP could be classified as a weak skin sensitiser. According to report WHO (Microsoft Word - Silver water disinfection and toxicity_2014V2.docx (who.int)) silver may have toxic effects via ingestion but in high concentrations (at least more than 100 mg/kg).

Ref

  1. Ferdous, Z.; Nemmar, A. Health Impact of Silver Nanoparticles : A Review of the Biodistribution and Toxicity Following Various Routes of Exposure. Int. J. Mol. Sci. 202021, 2375, doi:doi:10.3390/ijms21072375.
  2. Katsumiti, A.; Gilliland, D.; Arostegui, I.; Cajaraville, M.P. Mechanisms of Toxicity of Ag Nanoparticles in Comparison to Bulk and Ionic Ag on Mussel Hemocytes and Gill Cells. 2015, 1–30, doi:10.1371/journal.pone.0129039.
  3. Lekamge, S.; Miranda, A.F.; Abraham, A.; Li, V.; Shukla, R.; Bansal, V.; Nugegoda, D. The Toxicity of Silver Nanoparticles ( AgNPs ) to Three Freshwater Invertebrates With Different Life Strategies : Hydra vulgaris , Daphnia carinata , and Paratya australiensis. Front. Environ. Sci. 20186, 152–165, doi:10.3389/fenvs.2018.00152.
  4. Moon, T. Comparison of toxicity of uncoated and coated silver nanoparticles. J. Phys. Conf. Ser. 201349, 012025, doi:10.1088/1742-6596/429/1/012025.
  5. Xiu, Z.; Zhang, Q.; Puppala, H.L.; Colvin, V.L.; Alvarez, P.J.J. Negligible Particle-Specific Antibacterial Activity of Silver Nanoparticles. Nano Lett. 201212, 4271–4275.
  6. Mitra, C.; Gummadidala, P.M.; Afshinnia, K.; Corrin, R.; Baalousha, M.A.; Lead, J.R.; Chanda, A. Citrate-coated silver nanoparticles growth- independently inhibit aflatoxin synthesis in Aspergillus parasiticus. Environ. Sci. Technol 201751, 8085–8093.
  7. Melnikova, N.; Knyazev, A.; Nikolskiy, V.; Peretyagin, P.; Belyaeva, K.; Nazarova, N.; Liyaskina, E.; Malygina, D.; Revin, V. Wound Healing Composite Materials of Bacterial Cellulose and Zinc Oxide Nanoparticles with Immobilized Betulin Diphosphate. Nanomaterials 202111, 713.
  8. Alexandra, C.; Rayyif, S.M.I.; Mohammed, H.B.; Curut, C.; Chifiriuc, M.C.; Mih, G.; Holban, A.M. ZnO Nanoparticles-Modified Dressings to Inhibit Wound Pathogens. Materials (Basel). 202114, 3084.
  9. Webster, T.J. Antimicrobial Double-Layer Wound Dressing Based on Chitosan / Polyvinyl Alcohol / Copper : In vitro and in vivo Assessment. Int. J. Nanomedicine 202116, 223–235.
  10. Lou, Z.; Han, X.; Liu, J.; Ma, Q.; Yan, H.; Yuan, C.; Yang, L.; Han, H.; Weng, F.; Li, Y. Nano-Fe3O4 / bamboo bundles / phenolic resin oriented recombination ternary composite with enhanced multiple functions. Compos. Part B 2021226, 109335, doi:10.1016/j.compositesb.2021.109335.
  11. Kim, J.S.; Song, K.S.; Sung, J.H.; Ryu, H.R.; Gil, B.; Cho, H.S.; Lee, J.K.; Yu, I.J. Genotoxicity , acute oral and dermal toxicity , eye and dermal irritation and corrosion and skin sensitisation evaluation of silver nanoparticles. Nanotoxicology 20137, 953–960, doi:10.3109/17435390.2012.676099.

Round 2

Reviewer 1 Report

Accept as it is.

Author Response

We are grateful to the reviewer for his valuable comments and for his kind help with the manuscript improvement. We will proceed as suggested.

Reviewer 2 Report

Dear Author, 

I think you don't understand this question: 

In your experiment in In which mechanism do wounds heal? By contraction or epithelization?

Author Response

First of all, I would like to apologize for answering in the unclear way, I am very sorry.

In our case mice healing process was observed in a complex way, but mostly epithelialization. Nevertheless some contraction was also observed.

The epithelialization, through the formation of granulation tissue is  clearly visible on histological sections in our Figure 11.

However some features of contraction was also visible. A contraction, since mice have an additional subcutaneous muscle layer - panniculus carnosus was observed.